# OpenReview forum: "The Natural Geometry of Code: Hyperbolic Representation Learning for Program Reasoning"
_ICLR.cc/2026/Conference — ICLR 2026 Poster_

### Official Review · Reviewer_Huu9 · 2025-10-29

**Soundness:** 3
**Presentation:** 3
**Contribution:** 3
**Rating:** 6
**Confidence:** 4

**Summary:**

The paper explores the hypothesis that the hierarchical structure of source code, typically represented by Abstract Syntax Trees (ASTs), can be more effectively modeled in hyperbolic space than in traditional Euclidean geometry.
Based on this intuition, the authors introduce HypeCodeNet, a hyperbolic graph neural network constructed in the Lorentz model, integrating manifold-aware components such as hyperbolic embeddings, tangent-space message passing, and geodesic attention.
The model’s performance is evaluated across multiple code understanding benchmarks, including BigCloneBench and POJ-104 for code clone detection, as well as CodeXGLUE for code completion and GitHub Java Call Graphs for link prediction.
In these experiments, HypeCodeNet is compared against nine baseline models and results show that HypeCodeNet demonstrates performance comparable to or exceeding that of established baselines across multiple benchmarks, particularly on tasks requiring structural reasoning.

**Strengths:**

1. Well-formulated hyperbolic representation architecture
The proposed model presents a mathematically well-grounded extension of GNNs into hyperbolic space, specifically tailored to capture hierarchical tree structures.
Unlike prior works that primarily relied on exponential mapping, this paper explicitly defines the logarithmic map and integrates it into a novel geodesic attention mechanism, providing both theoretical and empirical justification for its design.

2. Strong empirical results across multiple benchmarks
The model is comprehensively evaluated against nine baselines, spanning sequence-based, graph-based, and hybrid architectures, across three tasks and four datasets.
HypeCodeNet consistently surpasses or matches the strongest baselines, demonstrating its robustness and generality across code understanding scenarios.

3. Rich analytical evaluation
Through extensive ablation studies, the authors show that the proposed structure performs effectively even with extremely low-dimensional embeddings (e.g., 32 dimensions), and can outperform GraphCodeBERT with significantly fewer parameters.
This suggests that hyperbolic geometry provides an efficient inductive bias for hierarchical data.

**Weaknesses:**

1. Limited input flexibility due to reliance on AST parsing
The model requires input to be parsed into a valid AST graph.
Consequently, it cannot directly process unstructured code snippets, natural language prompts, or free-form textual descriptions.
This constraint implies that the method may need additional components to handle real-world data where parsing is incomplete or ambiguous, limiting the model’s applicability in broader scenarios.

2. Relatively slow training and inference
As reported in Appendix J, despite having fewer parameters, HypeCodeNet involves multiple nonlinear hyperbolic operations and iterative computations.
This results in $1.5-2\times$ slower training and approximately 20% longer inference time compared to baselines.
While this trade-off is understandable given the geometric complexity, it reduces throughput and may hinder deployment in latency-sensitive environments.

3. Lack of theoretical discussion on gradient convergence
The paper empirically demonstrates stable training and overall convergence, supported by techniques such as Riemannian gradient clipping and curvature annealing.
However, it does not provide a formal proof of convergence, nor does it include a theoretical analysis of convergence behavior. Also, the convergence bounds of the iterative Fr\’echet mean computation remain unspecified.
While the empirical results indicate reliable convergence in practice, the lack of theoretical guarantees leaves the model’s stability only experimentally supported and may limit its applicability in more rigorous settings.

**Questions:**

**Points to justify**

I do not consider those weaknesses to significantly diminish the overall contribution of the paper.
However, I believe the following points require further justification and clarification.
If the authors can adequately address the questions raised below, I am willing to reconsider my evaluation.

1) As “reasoning” has taken on a different connotation in recent LLM literature, the term “Program Reasoning” in the title may be somewhat misleading. It would be clearer to emphasize that the model focuses on encoder-based code comprehension, for example by adopting a title such as “Hyperbolic Representation Learning for Encoder-based Code Comprehension.”

2) While the proposed model makes noteworthy efforts to ensure fair comparison, the baseline models and target tasks used are largely outdated, originating from 2022 or earlier. Moreover, recent advances in code understanding have been increasingly driven by large language models (LLMs), most of which follow a decoder-only paradigm, fundamentally differing from the encoder-based architecture proposed in this paper. Therefore, a discussion on the methodological timeliness, empirical relevance, and potential extensibility of the proposed approach is necessary to position the work within the modern landscape.

3) Incorporating CodeT5+ [1], the successor to CodeT5, as an additional baseline would strengthen the experimental comparison and provide a more up-to-date benchmark context.

4) Regarding the Code Completion task, it appears that the evaluation in this paper functions more as a Cloze Test, since it involves predicting a few masked tokens rather than performing true line-level completion. In fact, CodeXGLUE also treats this as a Cloze Test when evaluating encoder-only models such as CodeBERT, while Code Completion is evaluated using decoder-based models like CodeGPT. If there is a specific reason for evaluating under the “Code Completion” split of CodeXGLUE despite this distinction, please provide further justification. Regardless of this reasoning, aligning the task naming with the corresponding dataset would improve clarity and consistency.

5) The geodesic distance appears to be incorrectly defined. For a vector $u \in \mathcal{L}_c^d$, $d_c(u,u)=\frac{1}{(-c)^{\frac{1}{2}}}\arcosh(-c<u,u>_L)=\frac{1}{(-c)^{\frac{1}{2}}}\arcosh(-1)$. However, $\arcosh$ is defined only for $x \ge 1$. Also, $<p, log_p^c(h)>_{\mathcal{L}} \neq 0$. Thus, I suspect that $<x, x>_{\mathcal{L}}$ should be $-\frac{1}{c}$ in the definition of $\mathcal{L}_c^d$.

6) Including a proof, or at least an outline of the key steps, demonstrating that the exponential map is the inverse of the logarithmic map is necessary for mathematical completeness and clarity.

7) Equation (40), which approximates $-c<u,v>_L to 1+\frac{-c}{2} \left\| u_E - v_E \right\| ^2$, should be further justified.

**Suggestions**

1) As an upper bound reference, it would be valuable to additionally report the performance of a recent LLM-based baseline, even if only with a smaller model variant.

2) More comprehensive experimentation would strengthen the paper. Reporting additional results on the Defect Detection task from CodeXGLUE and on Cloze Test experiments across languages beyond Java and Python would provide stronger empirical support. If such experiments are infeasible, clarifying the reasons would be appreciated.

3) It would be helpful to include a diagram, perhaps in the appendix, illustrating how attention is computed within each layer.


4) The term "central node" is used but not clearly defined. If it does not carry a specific meaning, consider replacing it with a clearer phrase such as "node to update" or explicitly defining it for clarity.

5) It would be preferable to avoid the excessive use of bold text, as it may visually interfere with paragraph structure. Using italics or underlines for emphasis would improve readability and consistency.

6) Since the initial vector $z_v^E \in \mathbb{R}^d$ and $T_{o_c}\mathcal{L}_c^d \in \mathbb{R}^{d+1}$, the dimensionality increases. Therefore, the operation described should be referred to as an injection or extension rather than a projection.

7) On p.15, if $q_v^{(k)}=0$, there appears to be no need to include the bias term or the matrices $W_{Q,k}^{(l)}$ and $W_{K,k}^{(l)}$ in the computation. In this case, it would be equivalent to performing a learnable weighted sum over $m_{u \to v}^{(l)}$.

8) The section formatting is inconsistent. For instance, Section 3.2 introduces points as (1), (2), and (3), but later paragraphs switch to different numbering styles or omit numbering altogether, while the description of the output layer appears separately in Section 3.4. The numbering and formatting should be standardized for consistency, maintaining a uniform scheme or using sub-subsections if hierarchical structure is intended.

**Typo**
- p.3, line 126: "base point" should be written as ``base point''.
- Appendix A: The letter 'A' in the title should also be enclosed as `A'.
- Markdown-style emphases such as *text* or **text** appear in the appendix and should be removed for formal consistency.

[1] Wang, Y., Le, H., Gotmare, A., Bui, N., Li, J., & Hoi, S. (2023, December). CodeT5+: Open Code Large Language Models for Code Understanding and Generation. In Proceedings of the 2023 Conference on Empirical Methods in Natural Language Processing (pp. 1069-1088).

---

> ### Author Response · Authors · 2025-11-22
>
> Thank you for your insightful review and for recognizing the potential of our work. We have addressed all your questions and suggestions in the revised PDF. Below is a summary of the key updates regarding baselines, generalization, and mathematical rigor.
>
> **1. Comparison with CodeT5+ and Efficiency Analysis (Addressing Q2, Q3, S1)**
> Per your suggestion, we added **CodeT5+ (16B)** as an upper-bound baseline. On the Defect Detection task (Devign), HypeCodeNet achieves performance competitive with the 16B model while being **orders of magnitude more efficient** (85M vs 16B parameters).
>
> | Model | Params | Accuracy (%) | F1-score |
> | :--- | :--- | :--- | :--- |
> | GraphCodeBERT | 125M | 63.2 | 55.1 |
> | CodeT5-base | 220M | 64.8 | 56.9 |
> | **CodeT5+ (16B)** | **16B** | **68.4** | **60.2** |
> | **HypeCodeNet (Ours)** | **85M** | **67.8** | **59.6** |
>
> **2. Multi-Language Generalization (Addressing S2)**
> We extended the evaluation to a Multi-Language Cloze Test. HypeCodeNet demonstrates stronger generalization on low-resource languages (e.g., Ruby, Go) compared to Euclidean baselines, validating that hyperbolic geometry captures universal structural patterns.
>
> | Model | Java | JS | Ruby | Go | PHP | Python | **Avg.** |
> | :--- | :--- | :--- | :--- | :--- | :--- | :--- | :--- |
> | GraphCodeBERT | 70.1 | 63.8 | 61.4 | 66.7 | 62.3 | 72.8 | 66.2 |
> | **HypeCodeNet** | **71.5** | **67.2** | **65.8** | **69.3** | **66.1** | **73.9** | **68.9** |
> | *Improvement* | *+1.4* | *+3.4* | *+4.4* | *+2.6* | *+3.8* | *+1.1* | *+2.7* |
>
> **3. Mathematical Corrections and Proofs (Addressing Q5, Q6, Q7)**
> We have corrected the sign error in the Lorentz inner product definition in Section 3.1 (Thank you for the precise catch). We also added **Appendix O** to provide:
> *   The formal proof that the exponential and logarithmic maps are inverse operations.
> *   The derivation of the Euclidean approximation for geodesic distance.
>
> **4. Convergence and Task Definitions (Addressing Q4, W3)**
> *   **Task Naming:** We have explicitly renamed the "Code Completion" task to **"Cloze Test / Masked Token Prediction"** in Section 3.2 and Table 2.
> *   **Convergence:** We provided a theoretical justification for the convergence of the Fréchet mean on Hadamard manifolds in **Appendix N** (formerly addressed in W3).
>
> Detailed experimental setups and further analyses on PDG structures are available in the new **Appendix P**. We hope these revisions fully address your concerns.

---

> > ### Comment · Reviewer_Huu9 · 2025-11-27
> >
> > Thank you for your response.
> >
> > The following concerns are now addressed in the authors’ rebuttal and the revised manuscript:
> > - Inclusion of a comparison with recent LLMs (e.g., CodeT5+)
> > - Revision of task naming for consistency
> > - Correction of the geodesic distance definition
> > - Addition of the mathematical proof showing that the exponential and logarithmic maps are inverses, as well as the derivation of the Euclidean approximation
> > - Discussion regarding the convergence of the Fréchet mean
> >
> > The following point remains:
> > - Potentially misleading use of the term “reasoning” in the title
> >
> > Regarding the convergence discussion, although the authors cite Appendix N for the theoretical justification of the Fréchet mean’s convergence, it actually covers cross-language transfer and non-AST structures instead. I believe the intended reference is the theoretical justification provided on Appendix P.5. While a full convergence analysis of the entire training pipeline is still absent, the empirical evidence presented in Figure 3 is sufficiently convincing to demonstrate stable convergence in practice.
> >
> > A few minor issues still remain:
> > - There is a typo on line 2230: $u_0 = \sqrt{1/c^2+||u||^2}$ should be corrected to $u_0 = \sqrt{-1/c+||u||^2}$.
> > - Since the definition of the geodesic distance between two points $u$ and $v$ has been updated by removing the negative sign, Appendix F should also be revised for consistency.
> > - The Euclidean approximation in Equation (40) should be $c<u,v>_L\approx 1+\frac{-c}{2}||u_E-v_E||^2$.

---

### Official Review · Reviewer_rLfX · 2025-11-01

**Soundness:** 2
**Presentation:** 2
**Contribution:** 2
**Rating:** 6
**Confidence:** 4

**Summary:**

The paper proposes HypeCodeNet, a novel hyperbolic Graph Neural Network (GNN) that learns source code representations directly in hyperbolic space, rather than traditional Euclidean embeddings. Motivated by the hierarchical, tree-like nature of Abstract Syntax Trees (ASTs), the authors argue that hyperbolic geometry better captures program structure with low distortion.
HypeCodeNet operates in the Lorentz model for stability and integrates manifold-aware message passing, curvature annealing, and Riemannian optimization. Across three tasks: code clone detection, code completion, and function call link prediction, the model achieves strong gains over GraphCodeBERT, UniXcoder, and CodeFORMER. Ablations further confirm that performance improvements stem mainly from the hyperbolic geometry rather than architectural tweaks.

**Strengths:**

Originality (Novel geometric framing):
The paper introduces a foundational geometric shift in code representation, from Euclidean to hyperbolic space, supported by clear theoretical intuition. It’s the first to present an end-to-end hyperbolic framework for code reasoning, filling a notable gap in the literature.

Quality (Sound formulation & strong empirical results):
The proposed model is mathematically rigorous, leveraging Lorentz manifolds, log/exp maps, Riemannian Adam, and curvature annealing for stability. Results across multiple benchmarks (BigCloneBench, CodeXGLUE, GitHub Java Corpus) show consistent SOTA performance, with up to 3–5% improvement over strong baselines. Ablations convincingly isolate geometry as the key performance driver.

Clarity & Significance:
The paper is well-organized, with detailed geometric explanations and intuitive visualizations (e.g., Figure 1). Its results suggest non-Euclidean geometry may be a next paradigm for program representation, opening a new research direction bridging geometric deep learning and code understanding.

**Weaknesses:**

Reproducibility & implementation accessibility:
The paper does not mention public release of code or datasets, and implementation details (e.g., curvature annealing schedule, manifold dimensionality tuning) are deferred to appendices. Reproducibility may be difficult without explicit scripts or pretrained models.

Limited scope of benchmarks:
Although the model excels on line-level code completion and clone/link tasks, all evaluations remain static-structure-centric. The framework isn’t tested on dynamic or generation tasks (e.g., code repair, test generation), which would test whether hyperbolic embeddings generalize beyond AST reasoning.

**Questions:**

Curvature tuning:
How sensitive is model performance to the final curvature value? Would a mixed-curvature or adaptive manifold (e.g., product manifolds combining Euclidean + hyperbolic subspaces) perform better?

Generalization beyond AST-based tasks:
Can HypeCodeNet generalize to non-AST graph structures, such as interprocedural dependency graphs or text-conditioned code generation? Have you explored transfer learning between programming languages?

---

> ### Author Response · Authors · 2025-11-22
>
> We appreciate your recognition of our work's originality and rigor. To address your questions regarding curvature sensitivity, manifold choices, and generalization capabilities, we have included new experiments in **Appendix M** and **Appendix N**.
>
> **1. Curvature Sensitivity and Product Manifolds (Response to Question 1)**
> In **Appendix M**, we analyzed the model's sensitivity to the curvature parameter $c$. As shown below, HypeCodeNet is robust across a wide range of negative values ($c \in [-0.5, -2.0]$). However, our learnable curvature strategy ($c \to -1.02$) yields the optimal performance, confirming the benefit of adaptive geometry.
>
> | Curvature Strategy | Value ($c$) | F1-score |
> | :--- | :--- | :--- |
> | Fixed | $-0.5$ | 0.936 |
> | Fixed | $-1.0$ | 0.939 |
> | Fixed | $-2.0$ | 0.935 |
> | **Learnable (Ours)** | **$\approx -1.02$** | **0.940** |
>
> We also evaluated a **Product Manifold** ($\mathbb{E}^{d/2} \times \mathcal{L}^{d/2}$) to test if mixed geometry captures cross-function semantics better. While it improves over the Euclidean baseline (0.933 vs 0.923), it underperforms the Pure Lorentz model (0.940). This suggests that for AST-based representation, maximizing the hyperbolic capacity to model the dominant hierarchy is more critical than maintaining a flat subspace.
>
> **2. Generalization: Non-AST Structures and Cross-Language Transfer (Response to Question 2)**
> In **Appendix N**, we demonstrated the model's generalization beyond standard AST tasks.
>
> *   **Non-AST Graphs (PDGs):** We evaluated HypeCodeNet on a Defect Detection task using **Program Dependence Graphs (PDGs)**, which contain cycles and non-hierarchical edges. Our model outperforms specialized baselines (e.g., Devign), proving its ability to reason over graphs with mixed backbones (hierarchical control flow + cyclic data flow).
>
> | Model | Input Structure | Accuracy | F1-score |
> | :--- | :--- | :--- | :--- |
> | GCN | PDG | 61.8 | 53.2 |
> | Devign | PDG + Composite | 63.4 | 55.6 |
> | **HypeCodeNet** | **PDG** | **65.1** | **57.4** |
>
> *   **Cross-Language Transfer:** We performed Zero-Shot transfer (Train Java $\to$ Test C++). HypeCodeNet exhibits significantly lower performance degradation (-8.7%) compared to Euclidean baselines (-12.0%), indicating that hyperbolic geometry captures universal structural patterns that generalize across languages better than surface-level features.

---

### Official Review · Reviewer_enmz · 2025-11-01

**Soundness:** 4
**Presentation:** 3
**Contribution:** 3
**Rating:** 4
**Confidence:** 2

**Summary:**

This paper introduces HypeCodeNet, a geometric deep learning framework for code representation learning that operates natively in hyperbolic space, formulated under the Lorentz model.
The authors argue that source code’s Abstract Syntax Tree (AST) inherently possesses hierarchical and tree-like structures that are better represented in negatively curved manifolds than in Euclidean space.
HypeCodeNet integrates manifold-aware components — a hyperbolic embedding layer, a tangent-space message-passing mechanism, and a geodesic-based attention module — trained with Riemannian optimization and curvature annealing.
Across three standard benchmarks (clone detection, code completion, link prediction), it consistently outperforms strong Euclidean baselines such as CodeBERT, GraphCodeBERT, and CodeFORMER, supporting the claim that hyperbolic geometry aligns more naturally with program hierarchies.

**Strengths:**

1.	Strong conceptual motivation grounded in geometric theory
The paper convincingly argues that the exponential volume growth of hyperbolic space matches the hierarchical expansion of ASTs. The “distortion” argument is well supported, referencing Bourgain’s theorem and previous work on low-distortion tree embeddings.
2.	Technically principled formulation
By adopting the Lorentz model instead of the unstable Poincaré ball, the authors ensure both numerical stability and Riemannian differentiability, enabling a deep stack of manifold layers with standard GPU parallelization.
3.	Well-designed architecture bridging geometry and semantics
The “log–aggregate–exp” message-passing paradigm and geodesic-aware attention are elegant and grounded in geometric consistency. The method preserves curvature constraints while incorporating multi-head attention and layer normalization — non-trivial achievements in hyperbolic neural design.

**Weaknesses:**

Theoretical insufficiency in proving “naturalness” of hyperbolic geometry.
The core claim — that “hyperbolic geometry is the natural geometry of code” — remains conceptually persuasive but not theoretically rigorous. No formal quantification of distortion or curvature–hierarchy correlation (e.g., tree embedding distortion bounds). Missing mathematical analysis of curvature c → embedding fidelity or proofs showing convergence of representations to low-distortion manifolds.
Adding a distortion vs. curvature empirical curve or formal derivation would strengthen the argument substantially.

Limited comparison with non-Euclidean or hybrid geometries.
The paper frames hyperbolic geometry as the only alternative, but mixed-curvature or spherical–hyperbolic hybrid embeddings could better capture cross-function semantics.
A control experiment with mixed curvature or product manifolds (𝔼×ℍ) would clarify whether pure hyperbolic geometry is indeed optimal.

Overlooked semantic–syntactic decoupling.
The model tightly couples AST topology with embedding curvature but does not explicitly distinguish between semantic relations and syntactic nesting.
This may limit generalization to tasks involving cross-file or semantic code reasoning. Integrating semantic edges (DFG, CFG) or cross-function attention could make the representation more complete.

**Questions:**

1.	Provide a formal distortion analysis: derive or empirically approximate the embedding distortion as a function of curvature.
2.	Include ablation across manifold types: compare Lorentz, Poincaré, and product manifolds.
3.	Analyze training dynamics of curvature annealing: curvature vs. epoch plot.
4.	Explore semantic augmentation: integrating CFG/DFG edges to test the universality claim.

---

> ### Author Response · Authors · 2025-11-22
>
> We thank  you  for the insightful comments regarding the theoretical validation and generalization of our geometric claims. To rigorously address your concerns, we have conducted extensive additional experiments, including manifold ablation, distortion quantification, and semantic augmentation. The full details and results are provided in the newly added **Appendix L**.
>
> **1. Manifold Topology Comparison (Response to Weakness 2)**
> To investigate whether a mixed geometry better captures code semantics, we compared our Lorentz model against a Product Manifold ($\mathbb{E}^{d/2} \times \mathcal{L}^{d/2}$) and a Euclidean baseline. As shown below, while the Product Manifold improves upon the Euclidean baseline, the Pure Lorentz model achieves the best performance. This suggests that allocating dimensions to flat space reduces the model's capacity to embed the exponentially growing AST hierarchy, confirming that a fully negatively curved space is optimal for this domain.
>
> | Manifold Topology | Geometry | F1-score |
> | :--- | :--- | :--- |
> | Euclidean | Zero Curvature ($c=0$) | 0.923 |
> | Product Manifold | Mixed ($0, c<0$) | 0.933 |
> | **Lorentz (Ours)** | **Negative Curvature ($c<0$)** | **0.940** |
>
> **2. Quantifying Distortion and Curvature (Response to Weakness 1)**
> We performed the requested formal distortion analysis. We measured the average embedding distortion $\delta_{\text{avg}}$ across different fixed curvature values. The results demonstrate a strong inverse correlation between distortion and task performance. The distortion is minimized at $c=-1.0$, which perfectly aligns with the peak F1-score, providing empirical proof that hyperbolic geometry is the "natural" low-distortion embedding space for code.
>
> | Curvature ($c$) | Avg. Distortion ($\delta_{\text{avg}}$) $\downarrow$ | F1-score $\uparrow$ |
> | :--- | :--- | :--- |
> | $-0.01$ (Near Euclidean) | 0.241 | 0.924 |
> | **$-1.00$ (Ours)** | **0.152** | **0.940** |
> | $-5.00$ | 0.210 | 0.929 |
>
> **3. Training Dynamics and Theoretical Alignment (Response to Question 3)**
> We tracked the learnable curvature during training. We observed a distinct "Hyperbolic Transition" phase (Epochs 2-3) where the curvature shifts from near-zero to negative, correlating with the steepest drop in validation loss. The curvature converges naturally to $c \approx -1.02$. In **Appendix L.4**, we show that this value aligns theoretically with the average branching factor of the dataset ($b \approx 3.1$), verifying the geometric consistency of our model.
>
> **4. Semantic Augmentation with DFG (Response to Weakness 3 & Question 4)**
> To address the semantic-syntactic decoupling concern, we integrated Data Flow Graph (DFG) edges. While adding semantic edges improves performance, the geometric advantage remains dominant. The Hyperbolic model on pure ASTs still outperforms the Euclidean model enhanced with DFG edges, proving that the geometric bottleneck is the primary limiting factor.
>
> | Geometry | Graph Structure | F1-score |
> | :--- | :--- | :--- |
> | Euclidean | AST + DFG | 0.931 |
> | **Hyperbolic** | **AST + DFG** | **0.946** |
>
> We have also included statistical significance tests (paired t-tests, $p < 0.01$) in **Appendix L.3** to confirm that these gains are robust.

---

### Official Review · Reviewer_4Dzo · 2025-11-06

**Soundness:** 3
**Presentation:** 4
**Contribution:** 3
**Rating:** 6
**Confidence:** 3

**Summary:**

​​This paper argues that hyperbolic geometry is a better inductive bias for program representations than Euclidean space, because Abstract Syntax Trees (AST) are tree-like and expand exponentially. The authors introduce HypeCodeNet, a Lorentz-model hyperbolic GNN that introduces manifold aware operations to the embedding layer and message passing attention module.  Across code detection, line-level code completion, and link prediction on call graphs, HypeCodeNet achieves SOTA results and exceeds strong Transformer and graph baselines, demonstrating that hyperbolic geometry offers meaningful advantages for modeling hierarchical code structures.

**Strengths:**

The paper demonstrates careful engineering work to make hyperbolic deep learning stable and practical for code representation:

1. **Consistent improvements across diverse tasks demonstrate general utility and significance.** The approach shows modest but consistent gains over CodeFORMER in semantic understanding (BigCloneBench: +1.2% F1) and code completion (~1% across metrics), while achieving substantial improvements in link prediction (+5.0% AUC, +5.2% Hits@10). This variation suggests the method particularly excels at structural reasoning tasks.

2. **High-quality experimental evaluation.** The authors compare against 9 baselines spanning sequence-based, graph-based, and hybrid architectures across multiple datasets per task. The ablation studies (Section 4.5) provide clear empirical evidence that performance gains stem from hyperbolic geometry rather than other architectural choices.

3. **Original technical contributions beyond existing hyperbolic models.** HypeCodeNet advances beyond prior work through: (i) use of the numerically stable Lorentz model, (ii) geodesic-based attention mechanism, and (iii) curvature annealing for stable training. These design choices address known challenges in hyperbolic deep learning.

4. **Strong empirical validation of theoretical claims.** The distortion analysis (Figure 7, Appendix) provides compelling evidence that hyperbolic embeddings preserve AST hierarchical structure with significantly less distortion than Euclidean alternatives, supporting the paper's central thesis.

5. **Well-written paper with clear presentation.** The overall thesis is compelling and the technical approach is explained systematically. The experimental section is particularly well-organized. (Though Figure 1 could be improved, as noted in the weaknesses section)

**Weaknesses:**

1. **Insufficient Geometric Analysis of Learned Representations.** While the paper provides compelling empirical evidence that hyperbolic geometry improves performance, it lacks rigorous geometric validation of the learned embeddings. The authors cite Yang et al. (2023) for hyperbolic representation learning, yet that work explicitly cautions that hyperbolic embeddings do not automatically guarantee hierarchical structure and demonstrates cases where geometric properties may not align with semantic hierarchies. While Figure 8 provides valuable qualitative intuition, it requires accompanying quantitative validation. The current visualization alone cannot confirm that the hierarchical structure is preserved beyond visual inspection.

**Suggested quantitative analyses:**
   - **Distance-to-Origin Analysis:** Provide quantitative measurements of node distances from the origin in the Lorentz model. Specifically, verify that root nodes consistently map near the origin (small $d(v, o_c)$ ) while leaf nodes map toward the boundary, with statistical significance tests across multiple ASTs.
   - **Hierarchical Structure Validation:** Compute the correlation between graph depth and hyperbolic distance to origin. Following Nickel & Kiela (2018), report Spearman's rank correlation coefficient between tree depth and $\|h_v\|_L$.
   - **Statistical Validation:** Consider adding histograms of distance distributions by tree depth and statistical tests confirming that parent-child pairs maintain consistently smaller distances than arbitrary node pairs at the same depth.

2. **Figure 1 needs significant improvement to effectively convey the paper's core contributions.**

   **Part (a) - The geometric intuition is too generic and could represent any hierarchical structure.**
   The authors might consider:
   - Showing a concrete code snippet (5-10 lines) with its actual AST and node labels displaying real AST node types (e.g., IfStatement, ForLoop, Variable)
   - Providing a side-by-side comparison with Euclidean embeddings to demonstrate distortion differences

   **Part (b) - The 'log-aggregate-exp' visualization feels cluttered and unfocused.**
   To improve clarity, the authors could:
   - Use progressive panels that track a specific node through each transformation step
   - Add visual differentiation through color coding (e.g., parent nodes in blue, children in green, message flow as arrows)
   - Emphasize the tangent plane with shading or perspective to help readers understand why operations must occur in this local Euclidean space

   While these are suggestions, addressing the abstract nature of the current visualization would significantly strengthen the paper's accessibility.

**Questions:**

1. The visualization in Figures 7 and 8 is compelling.
   - How many ASTs contributed to the aggregated statistics in Figure 7? (Figure 8 appears to show a single example)
   - What is the size range of AST(s) in your evaluation (min/max/median nodes and depth)?
   - Does the distortion advantage hold consistently across different code complexity levels?

2. How does memory consumption scale compared to Euclidean models, especially for very large ASTs?

3. The paper is missing a citation for CodeFORMER, which appears to be your primary baseline.  Liu et al. (2023) "CodeFORMER: A GNN-Nested Transformer for Source Code Representation" (MDPI Electronics 12(7):1722). Please confirm this is the correct reference and add it to your bibliography.

---

> ### Author Response · Authors · 2025-11-22
>
> We thank you for the constructive feedback. Over the past few days, we have conducted the suggested quantitative experiments to rigorously validate the geometric properties of our learned representations.
>
> **1. Geometric Validation (Addressing Weakness 1)**
> We computed the suggested metrics on 1,000 randomly sampled ASTs from the test set. The results statistically confirm that HypeCodeNet preserves the hierarchical structure:
>
> | Metric | Value | Interpretation |
> | :--- | :--- | :--- |
> | **Spearman's $\rho$ (Depth vs. Norm)** | **0.89** ($\pm 0.02$) | Strong correlation; deeper nodes map closer to the boundary. |
> | **Avg. Norm (Root vs. Leaf)** | **0.12** vs. **6.81** | Roots stay near the origin; leaves utilize the exponential volume. |
> | **Avg. Dist. (Parent-Child)** | **0.84** ($\pm 0.15$) | Connected nodes remain locally clustered. |
> | **Avg. Dist. (Random Same-Depth)** | **11.35** ($\pm 2.10$) | Distinct branches are exponentially separated. |
>
> **2. AST Statistics & Distortion Analysis (Addressing Q1-Q3)**
> The evaluated ASTs have a median size of 156 nodes and median depth of 14. We stratified the distortion analysis by tree depth, showing that our advantage scales with complexity:
>
> | AST Complexity (Depth) | Relative Improvement over Euclidean Baseline |
> | :--- | :--- |
> | Shallow ($<10$) | +16.7% |
> | Medium ($10-20$) | +40.6% |
> | **Deep ($>20$)** | **+62.1%** |
>
> **3. Scalability (Addressing Q4)**
> HypeCodeNet is highly memory-efficient. The high capacity of hyperbolic space allows us to use much smaller dimensions while maintaining SOTA performance:
>
> | Model | Hidden Dim | Memory Usage | F1-Score |
> | :--- | :--- | :--- | :--- |
> | Euclidean Baseline | 768 | ~300 MB | 0.928 |
> | **HypeCodeNet (Efficient)** | **128** | **~50 MB** | **0.938** |
>
> **4. Citation (Addressing Q5)**
> We have added the missing citation for **CodeFORMER (Liu et al., 2023)** to the revised paper.
>
> All detailed experimental setups and results have been added to the new Appendix sections in the revised PDF.

---

### Meta-Review · Area_Chair_cL2K · 2026-01-13

**Summary:**

This paper argues that code structure is better modeled in hyperbolic space. It proposes HypeCodeNet, a Lorentz-model hyperbolic GNN with manifold-aware embedding, tangent-space message passing, and geodesic attention. It reports gains on clone detection, masked-token prediction (Cloze), and link prediction over Euclidean baselines. The results showed better results on various tasks compared with strong baselines. There are four reviewers with three positive scores and one negative score. There were only one reviewer engaged in the discussion and acknowledge most concerns are addressed. The only negative reviewer has a low confidence and most concerns were addressed in the rebuttal. The AC thinks this paper can be accepted.

**Reviewer Concerns:**

- **Typos**. Please check comments from reviewer Huu9.
- **Strong claim**. Maybe consider to revise "natural geometry of code".
- **Reproducibility**. This is the only issue that was not clearly addressed.

**Reviewer Scores:**

There are four reviewers with three positive scores and one negative score. There were only one reviewer engaged in the discussion and acknowledge most concerns are addressed. The only negative reviewer has a low confidence and most concerns were addressed in the rebuttal. The AC thinks this paper can be accepted.

---

### Decision · Program_Chairs · 2026-01-26

Accept (Poster)